# Achieving 'Active' 30 Minute Cities: How Feasible Is It to Reach Work within 30 Minutes Using Active Transport Modes?

Alan Both * , Lucy Gunn, Carl Higgs , Melanie Davern , Afshin Jafari, Claire Boulange and Billie Giles-Corti

Healthy Liveable Cities Lab, Centre for Urban Research, RMIT University, Melbourne 3000, Australia; lucy.gunn@rmit.edu.au (L.G.); carl.higgs@rmit.edu.au (C.H.); melanie.davern@rmit.edu.au (M.D.); afshin.jafari@rmit.edu.au (A.J.); claire.boulange@rmit.edu.au (C.B.); billie.giles-corti@rmit.edu.au (B.G.-C.)
* Correspondence: alan.both@rmit.edu.au

**Abstract:** Confronted with rapid urbanization, population growth, traffic congestion, and climate change, there is growing interest in creating cities that support active transport modes including walking, cycling, or public transport. The '30 minute city', where employment is accessible within 30 min by active transport, is being pursued in some cities to reduce congestion and foster local living. This paper examines the spatial relationship between employment, the skills of residents, and transport opportunities, to answer three questions about Australia's 21 largest cities: (1) What percentage of workers currently commute to their workplace within 30 min? (2) If workers were to shift to an active transport mode, what percent could reach their current workplace within 30 min? and (3) If it were possible to relocate workers closer to their employment or relocate employment closer to their home, what percentage could reach work within 30 min by each mode? Active transport usage in Australia is low, with public transport, walking, and cycling making up 16.8%, 2.8%, and 1.1% respectively of workers' commutes. Cycling was found to have the most potential for achieving the 30 min city, with an estimated 29.5% of workers able to reach their current workplace were they to shift to cycling. This increased to 69.1% if workers were also willing and able to find a similar job closer to home, potentially reducing commuting by private motor vehicle from 79.3% to 30.9%.

**Keywords:** 30 minute city; active transport; commuting patterns; transport interventions; urban transportation; OD-matrix

## 1. Introduction

Cities are economic hubs, providing access to employment, services, and resources [1]. Whilst many factors such as gentrification [2,3], culture and context [4,5] influence city development, it is city planning and economic development policies that predominantly determine a city's structure and the location of employment [6]. This affects the time spent commuting and the extent to which residents use active forms of transport such as walking, cycling, and public transport [7]. Poorly planned cities foster longer commutes, traffic congestion, and inactive and unsustainable lifestyles, and as a consequence, expose residents to environmental stressors (e.g., air and noise pollution), while increasing health inequity via inequalities in access to resources [7,8]. Good city planning that anticipates and manages these issues [9–14] is therefore critical to achieving sustainable urban development [7,15]. By 2050, cities will house 68% of the world's population. They already generate 75% of energy-related greenhouse gas emissions [1]. The need for healthier, sustainable urban development and transportation is now a recognized priority globally [10,13,14,16,17]. Hence, there is growing interest in creating cities that reduce the need to travel, with amenities and employment accessible using active transport modes such as walking, cycling, and public transit [18].

Australia is a highly urbanized nation, with over three quarters of Australians living in 21 cities [19,20]. Land use and transport planning in Australia has favored car-oriented

development, as evidenced by the large, sprawling, low-density footprint of capital cities such as Sydney, Melbourne, Adelaide, and Perth [21]. Consequently, motor vehicle dependency is high, with only 7.3% of Australian households not owning a motor vehicle, and 51.1% owning two or more vehicles [20].

Between 2010 and 2015, the cost of traffic congestion in Australian capital cities grew from $12.8 billion to $16.5 billion, with estimates that this could exceed $30 billion by 2030 under a 'business as usual' model [22]. Six of the nation's 10 fastest-growing areas are in Melbourne, placing pressure on road and transport infrastructure and congestion. This has the potential to be exacerbated in the post-COVID-19 pandemic recovery, as people shift from public transport to private motor vehicles for fear of disease transmission [23]. Hence, shifting the balance from a private motor vehicle-dependent city to a city that promotes sustainable and active transport is important, but not without challenge.

### 1.1. Private Motor Vehicle Useage and Active Transportation

In 2016, 9.2 million Australians worked outside the home and 79% commuted to work by private motor vehicle either as a driver or passenger, while only 5.2% either walked or cycled to work as their main transport mode [24]. On average, Australians traveled 16.5 km to reach their workplace [24]. Nevertheless, for many Australian city dwellers, some travel by private motor vehicle appears to be a preference rather than a necessity [25]. For example, a 2017 national Australian livability study found that, although 24–44% of employed workers over 15 years of age live and work in the same area, only 3–11% commuted by walking or cycling [21]. These studies suggest that there is an opportunity to encourage active transport when commuters live in proximity to important destinations.

### 1.2. Benefits of Active Transportation

Walking, cycling, and public transport as active forms of transport induce physical activity. Physical activity has mental and physical health benefits in terms of mitigating and reducing non-communicable diseases such as diabetes type 2, ischaemic heart disease and stroke, and some cancers including uterine and breast cancers among others [1,6–8]. Active transport is sustainable and equitable, with walking and cycling contributing no emissions whilst public transport is able to transport large numbers of people more efficiently than single-occupancy vehicles [7,8]. For cities experiencing growth, uptake of active transport is integral for reducing the pressure on road networks and for improving network-wide travel times. Furthermore, as part of a broader strategy for city-wide planning and infrastructure, active transport plays a pivotal role because of these co-benefits in making cities healthy, sustainable and efficient. Active transport requires appropriate infrastructure, such as sidewalks, cycle paths or improved public transport service, yet it is unclear which of these modes would have priority in achieving city-wide policy and planning initiatives that link people to important destinations.

### 1.3. Using Policy Initiatives to Shift to Compact Mixed-Use Developments with Public Transport

Different models of urban development are needed to maintain the livability of rapidly growing and congested cities. Globally, some initiatives examine the concept of a 15 min neighborhood [26–28]; however, in Australia, the focus of this study, the Victorian Government proposes a city of '20 min neighborhoods' following similar initiatives to those in Portland, OR, USA [29] and Tempe, Arizona [30]. In the 20 min neighborhood ideal, residents have access to important destinations including essential shops, services and public transport within 800 m, offering a 20 min return walking trip from home [31]. However, the 20 min neighborhood does not include access to employment. Some argue that to relieve congestion in cities, we must reduce the need to travel by redistributing employment across cities [24,32]. Hence, as an alternative to Melbourne's 20 min neighborhood, the New South Wales Government proposes a '30 min city' where most people have employment and amenities accessible within a 30 min walk or public transport trip [33]. Achieving the 30 min city has the potential to increase access to both local employment and local

amenities, and hence would help create a more equitable city [34]. It would also align with human behavior and an empirically demonstrated preference for a 30 min travel-time budget [18].

### 1.4. Bringing Jobs Closer to Where People Live

Levinson [18] argues that two strategies are required to achieve the 30 min city: creating new jobs in housing-rich areas and creating new housing in jobs-rich areas. However, achieving local living and shorter commute times also requires behavior change, a diversity of accessible employment opportunities, and an alignment between employment opportunities and the skills and expertise of residents. Understanding the spatial relationships between the location of employment, the skills of residents, and transport opportunities are therefore crucial.

### 1.5. Evidence-Informed Planning and the Use of Accessibility Indicators

One way of understanding spatial variation in access to employment is to create spatial indicators of access. Spatial indicators are measures designed to support evidence-informed planning that can benchmark and monitor planning interventions over time, inform policy development and urban governance, and be used in community engagement to support democracy and sustainability [35,36]. At the global level, indicator frameworks have been proposed to assess both the UN's Sustainable Development Goals [37] and its New Urban Agenda [38], and include a selection of indicators that, if achieved, would support sustainable development in cities [39]. Mapped spatial indicators also assist in measuring the extent of policy implementation and infrastructure delivery [21,31,40,41], and could help to identify spatial equalities in access to amenities within cities [42].

Accessibility is influenced by the spatial distribution and quantity of destinations in an area, by the amenities co-located with those destinations [43,44], and also by the quality, quantity and type of transportation options [45]. Accessibility indicators are often used to examine the relationship between land use and transport planning. Geurs and Eck (2001) define accessibility indicators as measuring 'the extent to which the land-use transport system enables (groups of) individuals or goods to reach activities or destinations by means of a (combination) of transport mode(s)' (p. 36) [46].

Many methods exist for measuring accessibility (see [47]). Briefly, accessibility can be measured by access to key destinations or land use categories, e.g., health, education, retail, employment, and banking within a certain time period by different transportation modes [30,47,48]. For example, the Structural Accessibility Layer (SAL) considers both the quantity and variety of destinations (including employment) accessible within a set time limit by transport mode [45], while the Metropolitan Accessibility Remoteness Index of Australia (Metro-ARIA [49,50]), primarily focuses on transport accessibility to common destinations such as education, health, shopping, public transport, and financial and postal services [50]. The Spatial Network Analysis of Public Transport Accessibility (SNAPTA) factors in destination accessibility and time hindrance, but focuses on access to the central business district, employment, retail, education opportunities, health, and leisure and recreational opportunities [51]. Other tools examining accessibility by public transit include the Spatial Network Analysis for Multimodal Urban Transport Systems (SNAMUTS [52–54]), and the Land Use & Public Transport Accessibility Index (LUPTAI) Tool [49]. Measures of access to public transport [21] and the numbers of jobs accessible within 30 min by different transport modes [55], are examples of accessibility indicators that combine transportation mode and employment availability that could inform city and economic planners.

However, to date, these transport and land-use accessibility indicators have captured network accessibility and population location relative to areas of high amenity, where amenity is derived from the quantity and variety of destinations accessible in any given area. They have not accounted for the location of specific destination types or the specific skills and employment needs of local residents, relative to the jobs that may be available. As such, they only serve as general measures for access to employment.

*1.6. Potential Indicators to Measure Access to Employment and the 30 Minute City*

To assess the potential to create a 30 min city, it is important not only to measure the relative locations of jobs and workers, but also to account for the accessibility of employment using various types of transportation infrastructure (e.g., cycle paths, roads, and rail). The infrastructure for these transport modes typically varies spatially throughout most cities, impacting travel times; furthermore, it is also important to consider the local employment needs of residents. The large variety of employment measures appearing in the literature reviewed in the previous section [21,43,45,49,53,54,56,57], typically ignore access to jobs relevant to the skills of local residents, and consequently, they may overrepresent employment accessibility and opportunities.

To address such shortfalls, this study used data from the 2016 Australian Census to explore three questions in Australia's largest 21 cities: (1) What percentage of people currently commute to their work within 30 min by walking, cycling, public transport or driving? (i.e., baseline); (2) If all those within an accessible catchment shifted to an alternative mode, what percent of people could reach their current workplace within 30 min by each transportation mode? (i.e., mode shift); and (3) If it were possible for workers to take a similar job closer to home, or to move closer to their employment, what percentage of people could get to work within 30 min for each transport mode (i.e., job-worker shift). The aims were to identify baseline levels of accessibility to employment, and to identify which of these scenarios would be more effective in allowing people to access work within 30 min by active transport. This research aimed to provide evidence on the level of change in accessibility to employment that can be expected for each scenario and, by doing so, to provide evidence for city planning about how the redistribution of employment across cities fits within a 30 min city framework.

## 2. Method

*2.1. Data and Method Used to Create the Indicators*

The geographic extents of Australia's 21 largest cities were based on the boundaries as defined by the Australian Bureau of Statistics (ABS), with the Greater Capital City Statistical Areas (GCCSA) and the Significant Urban Area (SUA) representing capital and regional cities respectively. The ABS Statistical Area 1 (SA1) regions that comprise these regions then served as the basis for all further analysis.

*2.2. Baseline and Mode Shift*

The 2016 ABS Distance to Work (DTWP) census item [19] records commuting distance in kilometers grouped into categories, but not the duration, nor time of day at which traveling took place. As this study involved analysis of traveling to work within 30 min, it was necessary to infer travel speeds for walking, cycling, public transport use, and driving. The average journey-to-work speed for each travel mode was estimated using the Victorian Integrated Survey of Transport and Activity (VISTA [58]), based on trip time and distance. VISTA 2012–16 is a one-day household travel survey administered across Melbourne where participants self-report their address and all trips including geocoded origin and destination, primary travel mode, trip purpose, and departure and arrival times. For the purposes of this work, public transport includes trips that use a train, tram, or bus as the primary travel mode. All trip components were considered in the analysis, so as to include any walking, cycling, or driving required to complete the journey.

Using the average speed for each travel mode, the distance traveled in 30 min was estimated for each of the aforementioned distance-based categories. For each mode, the distances were then grouped into two categories: the number of workers traveling distances less than 30 min, and those traveling for greater than 30 min. The baseline was then recorded as the number of workers who could currently travel to reach their job within 30 min for each transport mode. Mode shift was recorded as the number of commuters who could reach their job via each mode, regardless of the mode they actually used.

*2.3. Job-Worker Shift*

The job-worker shift scenario assumes that the number and location of workers and jobs remain the same for each job category. It allows workers to exchange their job with a similar one, so that the workforce overall spends the least amount of time commuting. The purpose of this scenario was to investigate where workers could decrease their commute time, either by taking a job that is closer to home, or by moving closer to their job.

To create an indicator to measure this scenario, 2016 ABS Census data was used to identify workers' occupations and the location of jobs. Industry of Employment (INDP) codes, based on the Australian and New Zealand Standard Industrial Classification (ANZSIC) [59], were used to match workers' industry of occupation to relevant employment Destination Zones (DZNs [60]). When matching workers' homes to employment destinations, it was assumed that jobs with the same ANZSIC code were interchangeable. This would give workers enough leeway to exchange jobs, but only in cases where the jobs were similar.

The population-weighted centroids of SA1 areas were used to represent home locations and the centroids of destination zones were used as job locations. These locations were then snapped to the nearest non-highway road in the OpenStreetMap dataset [61]. Trip times were then calculated using a combination of OpenStreetMap and General Transit Feed Specification (GTFS) data [62]. GTFS data were sourced for each Australian state transit agency in September 2019.

Multi-modal origin-destination analyses from SA1s to DZNs were conducted using OpenTripPlanner [63], with a departure time of 7:45 a.m., a maximum travel time of three hours and a maximum walking distance of 100 km. While only the trips within 30 min are of interest, these large travel times and walk distances were chosen in order to ensure that OpenTripPlanner would produce complete trips for all possible commutes. Modes considered were walking, cycling, driving, and a combination of walking and public transport.

Commuting between cities is a common occurrence in Australia, as many of Australia's cities are located close together. To account for this, as well as any potential commuters outside the cities, the cities were buffered by 40 km to create compound city regions that could be analyzed as contiguous regions. This distance was chosen as it is the maximum distance reachable within 30 min by a car traveling at 80 km/h. It is important to note that, while the SA1 and destination zones within this buffer were also used within the analysis, the results presented are only based on data from within the city regions.

The Australian Census data provides the number of jobs by ANZSIC job type for each SA1 and destination zone [64]. Hence, by combining these counts with journey duration estimates, it was possible to model an optimal transport plan between SA1 home locations and relevant job catchment zones. Specifically, we mapped people to jobs in a way that minimizes the total travel time of all workers. Optimal transport plans were calculated for each ANZSIC job type and transport mode combination to determine the percentage of people in each SA1 that could reach their job within 30 min—a category referred to as 'job-worker shift'.

### 3. Results

*3.1. Transport Modes Used for Commuting by Region*

Table 1 shows the total working population for each region alongside the proportion of workers for each transport mode. Regardless of region, commuting by driving was by far the most popular mode (79.3% overall), although considerably lower in Sydney (68.1%) than any other capital or regional city. Driving was most common in the regional city of Toowoomba, Queensland, with 96.3% of workers commuting to work by private motor vehicle.

Commuting by public transport (16.8% overall) tended to correlate with city size. Public transport commuting was higher in Australia's two largest (19.0% in Melbourne and 27.5% in Sydney) and medium-sized (14.2% in Brisbane and 12.7% in Perth) capital cities compared with smaller capital and regional cities with a working population of

less than 500,000. However, for a city with a small population, Darwin—the Northern Territory capital city—performed higher than similar-sized cities for public transport commuting (7.9%).

**Table 1.** Total working population for each region and the proportion of workers by transport mode.

| Region | Working Population | Percentage of Workers Commuting by Transport Mode | | | |
|---|---|---|---|---|---|
| | | Walk | Cycle | PT | Drive |
| **Australia** | | | | | |
| 21 cities | 6,890,287 | 2.8 | 1.1 | 16.8 | 79.3 |
| **Capital Cities** | | | | | |
| Sydney, New South Wales | 1,854,673 | 3.8 | 0.7 | 27.5 | 68.1 |
| Melbourne, Victoria | 1,711,476 | 2.7 | 1.5 | 19.0 | 76.8 |
| Brisbane, Queensland | 854,231 | 2.5 | 1.1 | 14.2 | 82.1 |
| Perth, Western Australia | 728,450 | 1.8 | 1.0 | 12.7 | 84.6 |
| Adelaide, South Australia | 466,545 | 1.9 | 1.1 | 10.8 | 86.2 |
| Canberra, Australian Capital Territory | 166,704 | 3.7 | 2.8 | 8.5 | 85.1 |
| Hobart, Tasmania | 76,670 | 5.2 | 1.2 | 6.4 | 87.3 |
| Darwin, Northern Territory | 58,418 | 3.6 | 2.3 | 7.9 | 86.3 |
| **Large Regional Cities** | | | | | |
| Gold Coast, Queensland | 221,118 | 2.4 | 0.8 | 4.8 | 92.0 |
| Newcastle, New South Wales | 161,510 | 2.1 | 0.8 | 3.0 | 94.1 |
| Sunshine Coast, Queensland | 99,793 | 2.5 | 0.8 | 2.7 | 94.1 |
| Wollongong, New South Wales | 98,259 | 2.5 | 0.3 | 7.6 | 89.6 |
| Geelong, Victoria | 86,678 | 2.4 | 0.7 | 6.5 | 90.4 |
| Townsville, Queensland | 65,134 | 1.8 | 1.5 | 1.6 | 95.1 |
| Cairns, Queensland | 53,819 | 3.2 | 2.0 | 2.7 | 92.1 |
| Toowoomba, Queensland | 46,419 | 2.5 | 0.8 | 0.5 | 96.3 |
| Ballarat, Victoria | 34,431 | 2.8 | 0.7 | 3.7 | 92.8 |
| Bendigo, Victoria | 32,770 | 2.7 | 0.7 | 2.1 | 94.5 |
| Mackay, Queensland | 29,684 | 1.9 | 0.9 | 2.5 | 94.6 |
| Launceston, Tasmania | 28,771 | 4.3 | 0.6 | 1.7 | 93.5 |
| Albury Wodonga, Victoria | 14,734 | 3.6 | 0.6 | 0.4 | 95.3 |

In the 2016 Census, only 2.8% of Australian workers commuted to work by walking, and 1.1% by cycling (see Table 1). The highest levels of walking commuters were observed in two relatively small Tasmanian cities—Hobart and Launceston (5.2% and 4.3% respectively); levels were lowest in the low-density sprawling capital cities Perth (1.8%) and Adelaide (1.9%), and smaller regional cities in Queensland (Townsville (1.8%); and Mackay (1.9%)).

Regardless of region, cycling was the least popular mode, with the largest proportion of commuter cyclists observed in Canberra (2.8%), followed by Darwin (2.3%), and Cairns (2.0%). Despite the potential for cycling in smaller regional cities, cycling prevalence was lowest in the smaller regional New South Wales city of Wollongong (0.3%) and noticeably low in Sydney (0.7%), Australia's most populous capital city.

*3.2. Calibrating 30 Minute Commuting Times Using VISTA*

Table 2 shows that the walking and cycling speeds observed were close to that reported in the literature (i.e., 5 km/h and 13 km/h respectively), but public transport and driving was much slower than typical speeds for those vehicles, likely due to traffic congestion and peak travel impacts. Public transport and driving typically had slower average speeds for short trips, reflecting the use of local roads, and higher speeds for longer trips, reflecting train and highway use. To account for this, a distance-based linear regression model was fitted to estimate speed given the distance (d), with the results shown in Table 2.

**Table 2.** Average values for journey to work for VISTA travel survey participants.

| Mode | Time (mins) | Distance Traveled (km) | Speed (km/h) [1,2] | Distance (km) Traveled in 30 min |
|---|---|---|---|---|
| Walking | 12.11 | 0.86 | 4.54 | 2.27 |
| Cycling | 30.16 | 6.60 | 13.10 | 6.55 |
| Public transport | 59.32 | 20.09 | **0.51** d + **8.78** | 5.89 |
| Driving | 28.81 | 15.66 | **0.70** d + **20.17** | 15.48 |

[1] d = distance. [2] Results in bold are statistically significant at $p < 0.001$.

Table 2 also shows distances that commuters could be expected to travel in 30 min. Notably, this suggests that in 30 min, pedestrians could travel around 2.3 km, about one third of the distance that could be traveled by a cyclist. Hence, cycling commuting appeared to better fit the 30 min city active transport ideal, with workers able to travel up to 6.6 km in 30 min. Notably, cyclists could also travel further in 30 min than those traveling by public transport (5.89 km). The approximate concordance between the average distance actually traveled (6.6 km) and the distance that could be traveled in 30 min (6.55 km) by cycling suggests that the 30 min cycling city could be more achievable than focusing on a 30 min walking city.

Figure 1 shows the distribution of distances traveled to work by VISTA travel survey participants. As expected, people traveling longer distances select transportation modes better suited for that trip length, with the shortest trips generally covered by walking and the longest by driving. While driving dominates mode choice for trip distances greater than 1 km, Figure 1 also shows that there is substantial overlap in longer commuting distances traveled by cycling and public transport. This overlap suggests there may be opportunities to shift modes to a more active form of transport.

Compared with other modes, workers spent more time commuting when using public transport (i.e., one hour on average); however, public transport in the Australian urban context has been estimated to include between 8 and 33 min of walking per day [65]. Average driving time was approximately 30 min (although it ranged from 5 to 78 min within two standard deviations). This suggests that Melbourne commuters had access to employment based on a 30 min driving city rather than a 30 min city centered on active transport. In 30 min, drivers traveled approximately 15 km to reach work (Table 2).

*3.3. Examining the Time Distributions of the Indicators*

Figure 2 shows the time distributions of commuting trips up to a maximum of three hours. Because the baseline scenario represents the proportion of total workers that currently use each mode, the non-driving modes have a much smaller proportion. For the baseline scenario (Figure 2a), walking and cycling tapered off quickly, with very few commuting longer than 40 min, whereas public transport had most commuters traveling more than 30 min. This was also consistent with the one-hour average journey time for public transport among the VISTA survey participants. Driving had a slightly narrower spread, with most commuters traveling less than 30 min.

The mode shift (Figure 2b) scenario resulted in large increases in walking and cycling for less than 30 min. The large number of trips longer than 30 min would be unlikely to change. The public transport distribution retained its spread and shape while increasing in size, with most commuters again traveling over 30 min. Driving appeared largely unchanged, which was expected given that most commuters already drive.

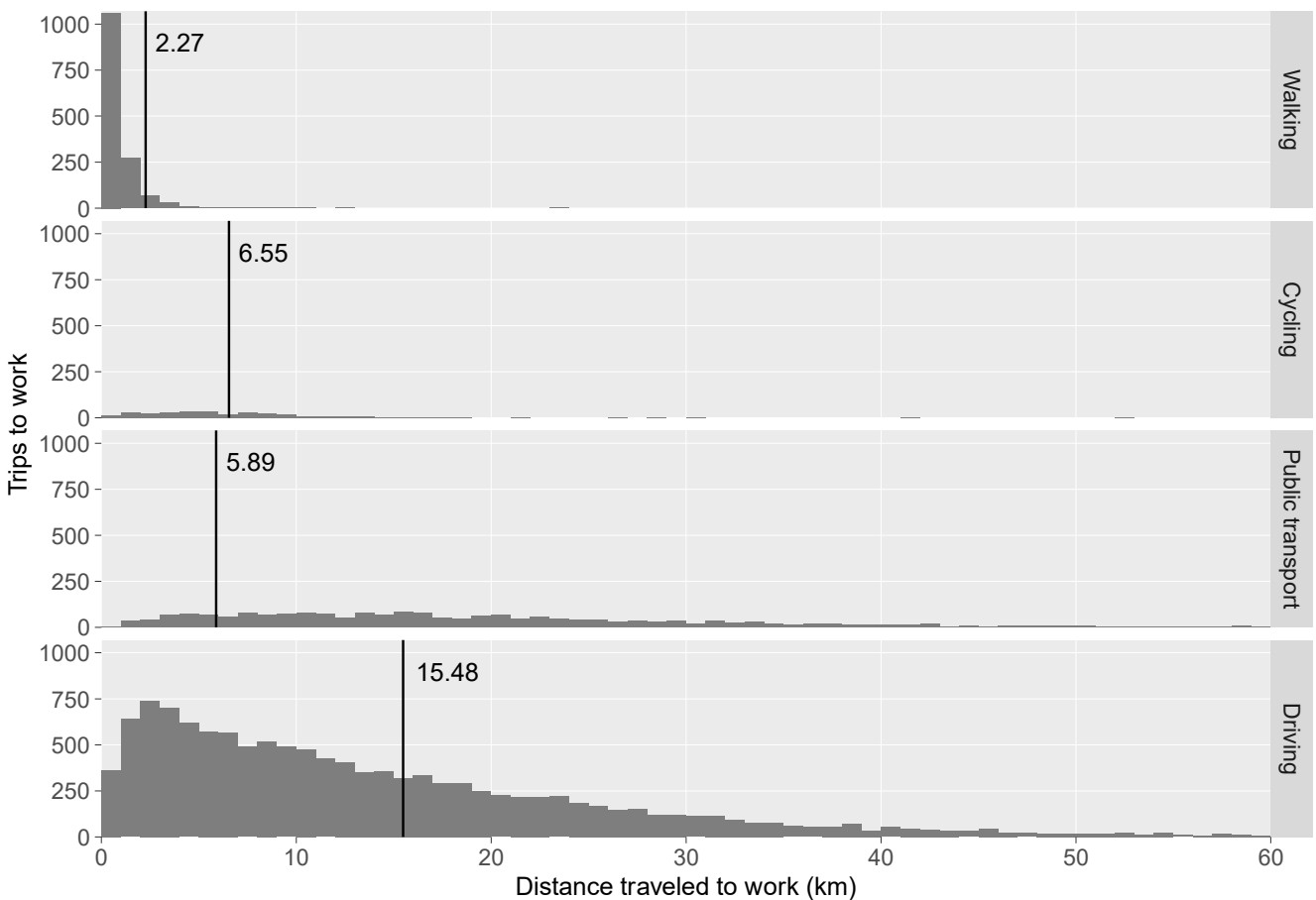

**Figure 1.** Distribution of distances traveled to work by mode for VISTA travel survey participants, with average distance traveled in 30 min marked by black vertical lines.

The job-worker shift scenario (Figure 2c) resulted in time distributions skewed towards zero for each mode, with the most extreme examples being for driving and cycling. For all modes except walking, the majority of commuters were able to reach work within 30 min, indicating the potential of this intervention for achieving a 30 min city.

*3.4. What Percent of People Could Reach Their Current Workplace within 30 min If They Shifted Mode?*

Figure 3 shows the proportion of workers who could reach their job within 30 min if they were to shift transport mode, with the baseline scenario showing the proportion of total workers that already do so. Overall, if workers were willing and able to change transport modes, an additional 5.8% of the Australian workforce could reach their jobs within 30 min if they shifted to walking. However, shifting to cycling or public transport appeared to enable the largest number of workers to reach their jobs within 30 min using an active mode. An additional 28.6% of workers could reach their job within 30 min if they switched to cycling, an additional 23.7% if they shifted to public transport and an additional 14.3% for driving. However, both the public transport and driving results should be interpreted with some caution due to limitations in our methods, which are considered in the limitations section.

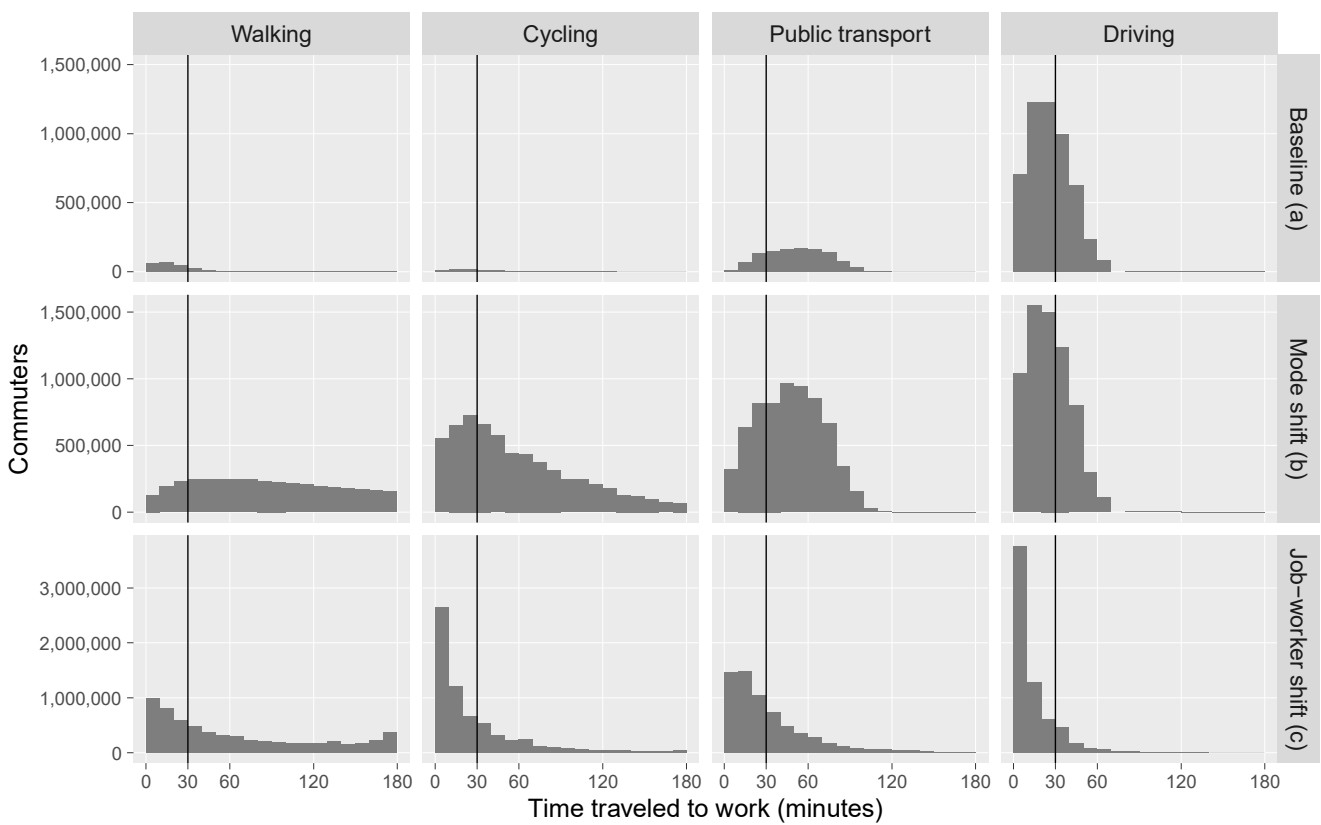

**Figure 2.** Distribution of time traveled to work by mode for commuters across 21 Australian cities.

*3.5. What Percentage of People Could Reach Work within 30 min If They Could Shift Jobs or Moved Closer to Their Work?*

The job-worker shift scenario assumes that all workers can adopt a particular travel mode and can change either their job or home location to minimize commuting time. Similarly to the mode shift scenario, all the active transport modes do well in the job-worker shift scenario, though cycling, public transport, and walking gained an additional 39.6%, 33.8% and 27.8% of the population, respectively, over the mode shift scenario (see Figure 3).

In the job-worker shift scenario, an additional 33.6% of the workforce could reach their jobs within 30 min by walking if they switched to a similar job that was closer to their home; 68.3% more could reach work within 30 min by cycling; an additional 57.5% could do so by public transport, but only an additional 38.2% would be able to do so by driving.

In summary, cycling performed best out of the active transport modes, allowing an additional 28.6% and 68.3% of commuters to reach work within 30 min for the mode shift and job-worker shift scenarios respectively.

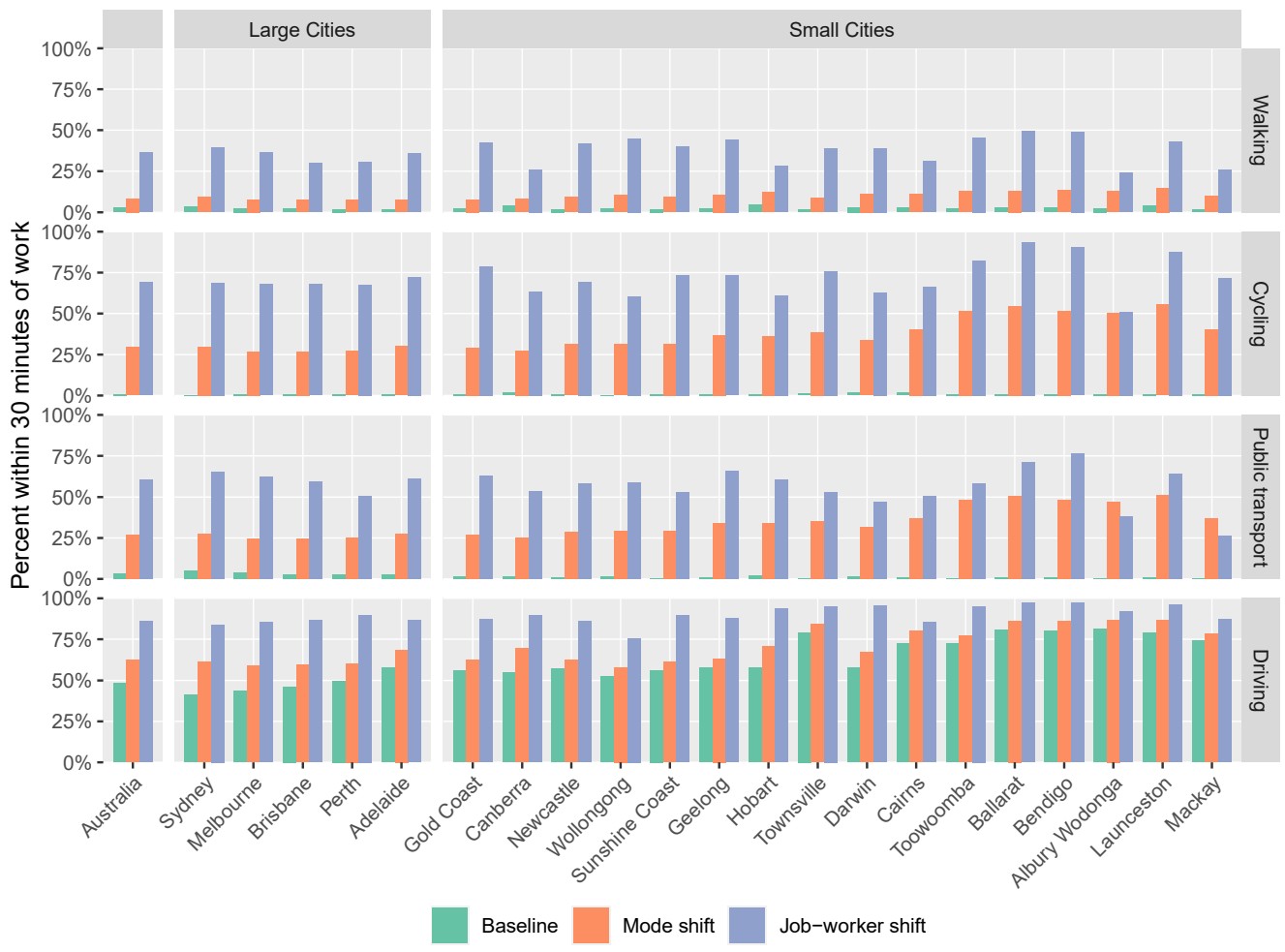

**Figure 3.** Percentage of commuters that can reach work within 30 min, or could if they shifted transport modes or changed jobs, overall and for cities ordered by total working population.

## 4. Discussion

Despite widespread recognition of the co-benefits to health and the environment of creating more compact cities with accessible amenities and employment that reduce the need for travel and encourage active modes of transport, there are challenges in doing so in cities traditionally designed to facilitate driving. We found that only 2.8% of Australian workers currently commute to work by walking, on average traveling approximately 0.86 km for about 12 min. This finding is consistent with similar research suggesting that 800 m is a walkable distance [66,67]. Based on empirically derived average walking speeds, we estimated that pedestrian commuters could walk around 2.25 km in 30 min; however, if all workers could be persuaded to shift modes, only 5.8% more workers could commute to work by walking within 30 min.

Currently, only 1.1% of Australian workers commute to work by bicycle. We estimated that (on average) these cyclists are currently cycling for approximately 30 min, and traveling just over 6.5 km. More Australian workers travel to work by public transport (16.8%) than by either walking or cycling, yet average distances currently traveled in 30 min in Melbourne (5.9 km) were actually slightly shorter than for cycling (6.55 km). Hence, while new greenfield developments are being developed as 20 min walkable neighborhoods with access to local services and amenities, in established areas in sprawling car dependent cities like those in Australia and North America, it may be more realistic to plan healthier, more sustainable cities based on 30 min cyclable (i.e., 6 km) or public transport (i.e., 10 km) distances. This would enable greater access to activity centers with employment, ameni-

ties, and mass rapid transport to city centers and other regional areas, helping to reduce inequalities in accessibility and improve country-wide development of sustainable cities.

To achieve the 30 min city aspiration, we explored two different scenarios: A mode shift scenario and a job-worker shift scenario.

### 4.1. Mode Shift

With the mode shift scenario, workers would shift from their current mode of transport to an active mode. This would require a multi-level intervention including behavior change, walking and cycling infrastructure investments, and enhanced public transport routes and services. Despite the considerable benefits of shifting to an active mode, we found that, even if all workers were able to switch to an active transport mode, if nothing else changed, 70.5% of workers would still have workplaces located further than 30 min away. Moreover, without demand management and disincentives to drive, some would choose to continue to drive, even if there was an alternative convenient mode [25]. In smaller capital and larger regional cities, the potential to shift workers to active modes was substantial. For example, Toowoomba in regional Queensland has a working population of around 46,000, and 51.6% of workers currently live within 30 min of work by an active mode. If willing and able, a large proportion of this population could shift to an active mode (i.e., walking, cycling, or public transport) if they had appropriate equipment and infrastructure.

A notable finding was that only 8.4% of workers could currently reach their workplace by walking if they chose to shift to this mode, while 27.1% could reach their current place of employment by public transport within 30 min. However, perhaps the least expected finding was that although only 1.1% of Australian workers currently commute by cycling, the mode shift results suggested that 29.5% of workers could reach their current workplace within 30 min if workers were willing and able to switch to cycling and had access to safe cycling infrastructure. This represents a significant opportunity in car-dependent cities and underscores the need for future investment in cycling infrastructure, which is considered further below.

### 4.2. Job-Worker Shift

The job-worker shift scenario would reduce the need to travel, by shifting either the location of relevant employment closer to workers' homes, or alternatively shifting workers closer to their place of employment. Compared with the mode shift scenario, it was estimated that the job-worker shift scenario would greatly increase the proportion of workers who could walk, cycle, or use public transport. However, the most significant finding under this scenario was that driving by private motor vehicle could be reduced from 79.3% to 30.9% if this shift could be implemented. In smaller capital cities and larger regional cities, the potential was even more substantial. For example, in the regional city of Ballarat, Victoria, where there are just over 34,000 workers, 93.6% of workers could cycle to work and 49.3% could walk under this scenario. However, implementing the job-worker shift scenario would require major policy reform involving land use, infrastructure, and employment locations to relocate jobs closer to people's homes [32].

### 4.3. Population Shifts to More Active Forms of Transport

Despite the co-benefits to health, the environment, and the economy of shifting to active modes of transport, achieving behavior change is challenging. There is growing policy focus on walkable communities and city planning that creates local communities with amenities (e.g., 20 min neighborhoods in Melbourne [31], 15 min neighborhoods in Paris [68]) or accessible employment within 30 min in Sydney) [33]. However, whilst city planning reforms are necessary, alone they are insufficient to change behavior. For example, Handy and colleagues argue that in the United States many people drive by choice, rather than out of necessity [25]. Similarly, Australian evidence finds that around one third of all trips less than 1 km, and 65% of those between 1 km and 1.9 km, involve driving by private

motor vehicle [69]. Hence, Handy, Weston and Mokhtarian [25] argue that city planners must explore ways to create accessible compact communities that reduce trip lengths.

To achieve this, multi-level interventions are required to shift populations to active transport in addition to city-planning interventions that create more compact pedestrian- and cycling-friendly neighborhoods as well as relocating employment closer to people's homes. Additional interventions include: (1) community education programs about the co-benefits of active transport and reduced vehicle emissions designed to change social norms; (2) cycling infrastructure that makes cycling safer and more efficient than driving, including end-of-trip facilities to increase the convenience of using active transport modes; (3) more frequent and direct public transport routes; (4) demand management strategies that reduce the convenience and competitiveness of driving (e.g., the availability and cost of park; road charging [6]); and (5) relocating jobs closer to residential locations. The latter would increase the job-housing balance in suburban development, thereby increasing access to local destinations, increasing active transportation trips [70], and helping to deliver the 20 min neighborhood [30,31].

However, as we have shown, one of the biggest opportunities to increase active transport could be achieved by encouraging more cycling. Figure 1 shows that there are many commuter car trips made within cyclable distances and the mode shift scenario suggests that over 26.7% of Melburnians are already within a cyclable distance of their job. Furthermore, a 2019 survey in Australia found that more than 57% of Australian households have access to one or more bicycles and 26% of respondents indicated that they were interested in starting cycling for transport purposes [71].

The importance of cycling as a mode of transportation has become evident in the context of the COVID-19 pandemic. For many people, working from home has reduced the need for travel and has emphasized the importance of local accessibility to shops and services, making the concepts such as the 20 min neighborhood and 30 min city ideal even more relevant. However, one of the most prominent observed travel behavior changes during the pandemic in many countries has been the shift away from public transport towards using private motor vehicles [23,72,73], with a recent Victorian survey suggesting that about 9% of public transport users plan to switch to cars post-pandemic [74]. These changes will have major impacts on traffic and traffic congestion, while also raising concerns about health effects due to the loss of physical activity and increased exposure to air and noise pollution [6,75].

Encouraging cycling for transportation provides a promising solution, given we have shown that, compared with public transport, it is competitive in terms of average distances traveled. However, to encourage more cycling [76], it is critical to make cycling safer, to minimize road trauma [77]. Moreover, to encourage more people to cycle more often, including women and vulnerable cyclists (e.g., children and older adults) [78,79], it will require investment in safer cycling infrastructure (separated cycle lanes and cycle paths) and behavior-change programs that change social norms [80–82].

*4.4. Strengths and Limitations*

In this study, we conducted a multi-modal origin-destination analysis between the population-weighted centroids of local neighborhoods (SA1s) and job centers (DZNs) using cross-sectional data. The use of cross-sectional data is a limitation, and precludes causal inferences being drawn. Collection of longitudinal data by data custodians would enable longitudinal analyses of the 30 min city for future work, and this is warranted. Furthermore, the use of centroids was a pragmatic compromise; while overlooking the local variation of a more complex address-level analysis (e.g., [83]), we opted to capture the broader macro variation in travel times across the cities, using an established method in the GIS literature [84,85]. Similarly, the use of OpenStreetMap was also a tradeoff; while new developments and fringe areas may be less complete, overall completeness for cities is very high [86], with the inclusion of cut-throughs and cycling tracks providing an advantage over official datasets [87].

To consistently capture public transit for each city, we used official GTFS data from state transportation agencies for our analyses. A strength of GTFS data is that it accounts for projected congestion in scheduled departures, although our OpenTripPlanner model did not account for congestion for the other modes. While it would be possible to mitigate the impact of congestion on public transit by selecting off-peak travel times, this was not done as services would be less frequent. The intention behind running all analyses during the morning peak was to ensure that the best possible travel times would be recorded to account for some temporal effects, regardless of travel congestion or public transit delays. However, methods for incorporation of spatiotemporal effects and data on traffic congestion for weighting travel times is an active field of research [88]. While this could provide a valuable extension to future work, it would require detailed congestion datasets that are of consistent quality and completeness across all cities.

Assumptions on commute duration were informed by the main mode of transport stated by participants of the VISTA survey. This may ignore possible trip-chaining including the use of park-and-ride facilities and is potentially subject to self-reporting biases; however, these are anticipated to be minimal, given that the VISTA survey relies on detailed travel diaries. However, we reason that the use of the main mode of transport is the larger and more relevant component of travel for the purpose of this study. Accommodating trip-chaining is left for future work. Additionally, we used linear regressions to model travel time based on distance and mean speed. This has limitations because it does not consider local variation in traffic congestion and its effects on travel time. In addition, we did not consider people working at different hours and the effects this will have on travel time. Future studies may wish to consider using a routing service to derive more realistic travel times for driving and public transport.

The generalization of our approach across cities may be less accurate for public transport and driving in the baseline and mode shift scenarios due to differences in the infrastructure and network of services available. This problem is highlighted in Ballarat, Victoria, where 3.7% of commuters take public transport and even fewer arrive within 30 min (0.7%). The mode shift scenario assumes that 50.2% could commute to their jobs by public transport, even though in reality, the public transport network in Ballarat may not currently support this shift. Notably, the job-worker shift measure for public transport does not have the same issues, as it utilizes the actual public transport network when calculating journey time. Nevertheless, the focus on Australia's 21 largest cities remains relevant for the planning of regional cities.

Finally, both the mode shift and the job-worker shift scenarios were intended to be aspirational 'what if' scenarios, where workers do not need to factor in the transactional costs (e.g., mental, social, financial) of moving house, getting a new job, or buying a bicycle. The approach we have taken has limitations: people do not change jobs often, and when they do, location is only one of many considerations. Furthermore, jobs may not be easily exchanged to be closer to peoples' homes and people may not be able to use active modes. Nevertheless, our scenarios show what form our society could take if we all prioritized living and working locally; choices that will be determined at least in part by land use, employment and regional economic development policies that determine the location and availability of employment across a city [6,32].

### 4.5. Where to Next?

To facilitate evidence-informed planning, the final set of indicators developed for this study will be disseminated through the Australian Urban Observatory (AUO) [89], a digital planning platform measuring livability across Australian cities. Indicator frameworks populated with administrative and spatial data are now being made available via digital platforms such as the AUO as a convenient resource to support knowledge translation, with the government, academia and practices providing evidence to assist policy development towards integrated, collaborative land use and transport planning [21,39] in linking to smart cities and sustainable development that resolves city-wide issues relating

to infrastructure, transport and governance [90–92]. In Australia, the AUO disseminates policy-relevant indicators available at varying degrees of disaggregation for Australia's 21 largest cities, where over three quarters of Australians currently live. These 21 cities are included in the Australian Federal Government's National Cities Performance Framework for cities [93], which aims to provide data ' . . . to help all levels of government, industry and the community to make the best policy and investment decisions'. The 30 min city indicators could inform decision-makers including local, state, and federal governments about where best to invest in walking and cycling infrastructure to maximize the number of workers who could reach their employment by active modes, or alternatively to identify areas that might benefit from relocating employment closer to workers' homes.

While relocating employment and building affordable housing closer to employment is a long-term strategy, this study's results provide policymakers with evidence to support increased investment in cycling infrastructure to make cycling more feasible and safer. Indeed, in Europe, investing in cycling infrastructure is recognized as a means to stimulate the economy post-COVID [94]. There is growing evidence on the cost-benefits of investing in cycling infrastructure. In reviewing 16 cycling studies, Cavill and colleagues found a benefit-cost-ratio as high as 1:32.5, whilst the median benefit-cost ratio showed $5 in benefits for every dollar spent on cycling infrastructure [95].

A good starting point would be prioritizing separate cycle paths on roads within 5 km of train stations, activity centers and high schools [76,96]; and creating opportunities for safe cycling within 2 km of primary schools [96,97]. However, infrastructure investments must be complemented by behavior-change programs designed to change social norms and increase the uptake of cycling.

In conclusion, the indicators developed in this study demonstrate that there are significant opportunities and benefits for cities seeking to create more accessible cities, through models such as the 30 min city. The urgency for this type of development has become even more prescient during the COVID-19 pandemic; and could be combined with the concept of the 20 min neighborhood designed to make key destinations required for daily living more proximate. However, critically, transitioning to cities that promote more cycling and public transport and reducing the need to travel by people living closer to their place of employment could address many contemporary problems associated with cities, including rapid urbanization, traffic congestion, air pollution, health and wellbeing; as well as reducing emissions and creating more sustainable and resilient cities. There is now growing evidence that this is viable, desirable, economically beneficial and would future-proof our cities.

**Author Contributions:** Conceptualization; Alan Both, Lucy Gunn, Melanie Davern, Claire Boulange and Billie Giles-Corti; Formal analysis, Alan Both and Carl Higgs; Funding acquisition, Billie Giles-Corti; Methodology, Alan Both, Lucy Gunn, Carl Higgs, Melanie Davern and Billie Giles-Corti; Project administration, Lucy Gunn and Billie Giles-Corti; Visualization, Alan Both; Writing—original draft, Alan Both, Lucy Gunn, Carl Higgs, Melanie Davern, Afshin Jafari, Claire Boulange and Billie Giles-Corti; Writing—review & editing, Alan Both, Lucy Gunn, Carl Higgs, Melanie Davern, Afshin Jafari, Claire Boulange and Billie Giles-Corti. All authors have read and agreed to the published version of the manuscript.

**Funding:** A.B.: L.G., B.G.C., C.B. and C.H. were supported by the NHMRC funded Australian Prevention Partnership Centre (#9100001). M.D. receives funding and the National Health and Medical Research Council MRF1200144. A.J. is supported by an Australian Government Research Training Program Scholarship.

**Institutional Review Board Statement:** Not applicable.

**Informed Consent Statement:** Not applicable.

**Data Availability Statement:** The data presented in this study and the code used to generate it are openly available at https://github.com/healthy-liveable-cities/30-minute-cities (accessed on 10 January 2021). The generalized OpenTripPlanner analysis code developed and used for the multi-modal accessibility analysis is available at https://github.com/healthy-liveable-cities/otp-multimodal-access-analysis (accessed on 10 January 2021).

**Acknowledgments:** The OpenTripPlanner implementation was adapted by Carl Higgs from original code authored Dhirendra Singh through the 'Decision making for lifetime affordable and tenable city housing' project under an ARC Linkage grant, LP130100008, and whose contribution of the code for this project we gratefully acknowledge.

**Conflicts of Interest:** The authors declare no conflict of interest.

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
