# Peer review of "Achieving ‘Active’ 30 Minute Cities: How Feasible Is It to Reach Work within 30 Minutes Using Active Transport Modes?"

_ijgi, doi:10.3390/ijgi11010058_

Round 1

Reviewer 1 Report

In this study, the authors assess the accessibility of workplaces in 21 Australian cities. In a baseline investigation, travel times are estimated. In a consecutive step, the potential for commuting trips < 30 minutes is calculated for four different modes (walking, cycling, PT, car) in two scenarios. Their analysis highlights the huge potential for commuting by bicycle in urban agglomerations. The study deals with a timely topic and adds another piece of evidence to the vast number of studies on this and related topics. However, the contribution to GIScience remains opaque (the applied method is standard GIS analysis), while the conclusions drawn from the study results rest on rather shaky pillars due to methodological shortcomings. I would therefore suggest thoroughly revising the manuscript and coming back with a stronger case. The following remarks might be helpful in this process:

  1. Introduction

The connection between urban morphology and structure on the one and commuting habits on the other hand is evident and well-grounded in the cited literature. However, “City planning and economic development policies” are not the only driving forces. There are definitely more factors, such as segregation, gentrification, cultural settings or simply inadequate or non-existing PT systems, just to name a few.

The argument for promoting sustainable transport systems in Australian cities is limited to congestion costs. I wonder whether negative environmental effects or public health play a role as well?

The title “Encouraging active forms of transportation” is not reflected in the subsequent paragraph. Please reconsider it in the revision.

Please either justify your restriction to North American and Australian cities in the list of reference approaches, or extend it to examples from the rest of the world (there is lot of recent work, such as https://doi.org/10.1371/journal.pone.0250080 and https://doi.org/10.3390/smartcities4010006). A critical reflection of the concept (such as https://doi.org/10.3390/su13020928) helps to embed the presented approach into a broader context.

I have two remarks to the introduction of accessibility indicators: First, you only consider spatial concepts of accessibility. There are definitely more accessibility concepts that are relevant in the context of this study. Please also consider for instance cultural attitudes towards different modes of transport, monetary costs, and health considerations (especially relevant in the current COVID-19 pandemic, where those who have the opportunity change from PT to private vehicles). Second, the temporal aspect is completely neglected. This is fundamental for assessing the accessibility with PT.

While the 15/whatever-minutes-city is a model that addresses urban planners and decision makers (provision of housing/working opportunities and transport infrastructure, accessibility), the two investigated scenarios only focus on individuals (mode shift, job shift). The latter is not only a function of available opportunities, but also even more of behavior change. This aspect is entirely ignored in the introduction to this study. The job shift scenario, as described in the method section, is a very hypothetical one. I would regard scenarios for attracting industry estates to available areas or building new neighborhoods as more relevant for decision makers.

  1. Method

The inference of travel time based on distance and mean speed is a very coarse estimation. Particularly in urban settings, travel time is not a linear function of distance for PT and car. I regard this aspect a major downside of the presented study. In addition, different working hours are not considered, although the time of departure has significant impact on travel times. I would recommend using routing services (Here, Google, etc. API) in order to achieve more realistic results. The retrospective fitting, as suggested in the results section, is only the second best solutions. In addition, reporting biases in self-reported travel times add another bias to the overall estimation in your model. This is particularly relevant for walking and PT trips (see e.g. https://doi.org/10.1111/sms.13636).

  1. Results

In remains unclear to which degree the modal split is a function of service quality. In order to be able to contextualize the results, the modal share for each mode should be related with PT service quality, share of bicycle ways and pedestrian infrastructure (sidewalks, pedestrian zones, traffic calmed areas etc.) respectively. In addition, land use and population density would be helpful variables with high explanatory value for mobility behavior (see e.g. reflection in https://doi.org/10.1016/S0140-6736(16)31578-1).

In the mode-shift scenario, intermodal trips are neglected. The combination of walking/cycling and PT extends the range of commuters substantially, especially if rail systems are considered. Please consider intermodal trips in your revision.

  1. Discussion

The discussion of behavior change is interesting and supports the conclusions of this study. However, this aspect is entirely missing in the introduction. Thus, the benefit of this particular study is hard to assess. Putting it into a (over-) simplistic question: “Is the low share of sustainable commuting mobility due to the result of car-centric spatial planning and development, or is it because of car-focused mobility behavior patterns?”

Reviewer 2 Report

The topic: Achieving ‘active’ 30-minute cities: How feasible is it to reach work within 30 minutes using active transport modes? presented herein is very important for both research and decision-making.

The paper generally is good with minor issues to improve.

The rationale for 30-minute cities has been not been extensively expounded in the paper. It has been explained but could be improved to boost the importance of this research. I suggest that you give a paragraph or two on the health benefits given walking, and cycling given that cycling has a good preference in this manuscript too.

I suggest you some reading to support the above assertion:

Bătăgan, L., 2011. Smart cities and sustainability models. Informatica Economică15(3), pp.80-

https://www.mdpi.com/2071-1050/11/7/2140

87.https://www.emerald.com/insight/content/doi/10.1108/ARLA-04-2017-

0133/full/htmlhttps://www.mdpi.com/2073-445X/10/8/870

On the organization of work, the work could be organized in journal format.

Also, the references and citations should be organized in the journal format in chronological order.

Statistical analysis with significancies would add more value to the discussion and importance of the paper's contribution.

The paper could be accepted after the suggested corrections

Reviewer 3 Report

This is a very well-written manuscript exploring the possibilities of achieving ‘active’ 30-minute cities in Australia. The manuscript presents a clear methodological approach that could be reproduced in other cities of the world (with modification regarding the sources of data), and moreover, explores different types of **-minute cities, e.g., including or not commuting, etc., as there are several schemes under which this concept is discussed in the literature and is implemented in different cities of the world.

I have a couple of comments and suggestions that might improve the manuscript:

  1. Line 8-9: I think writing that active mobility includes public transport is not correct. Public transport is a strong determinant of active mobility, but not active mobility itself. A better expression I found in the manuscript is to describe these three modes as modes of sustainable mobility.
  2. I think some more discussion is needed regarding the fact that, as I understand, the survey was based on the assumption that if a commuter could reach his destination in 30 minutes he/she would indeed shift modes. Of course, this is not always the case, but I understand that the research is about whether the cities could become 30-minute cities, not about whether indeed the modal split will change. If I understood that well, I think it needs some more discussion.
  3. Here and there, there are some spaces that are not needed: e.g., Line 526, Line 541, etc.
  4. I think you should avoid all capitals in the reference list.

Round 2

Reviewer 1 Report

The authors replied extensively to the reviewers' initial comments, which is very much appreciated.
I do not fully agree with how the authors solved the methodological shortcomings, but, at least, they included them in the discussion of limitations. In this sense, the study will serve as kick-start for further research.